# Direct Urine Resistance Detection Using VITEK 2

**DOI:** 10.3390/antibiotics11050663

**Published:** 2022-05-15

**Authors:** Eva Torres-Sangiao, Brais Lamas Rodriguez, María Cea Pájaro, Raquel Carracedo Montero, Noelia Parajó Pazos, Carlos García-Riestra

**Affiliations:** 1Grupo *Escherichia coli*, Fundación Instituto de InvestigaciónSanitaria (FIDIS), Hospital Clínico Universitario de Santiago de Compostela (CHUS), 15706 Santiago de Compostela, Spain; 2Clinical Microbiology Lab, University Hospital Marqués de Valdecilla, 39008 Santander, Spain; 3Instituto de Investigación Sanitaria Marqués de Valdecilla (IDIVAL), 39011 Santander, Spain; 4Dto Microbiology at Medical School, University of Santiago de Compostela, 15705 Santiago de Compostela, Spain; braislamasrodri@gmail.com; 5Clinical Microbiology Lab, Hospital Clínico Universitario de Santiago de Compostela (CHUS), 15706 Santiago de Compostela, Spain; maria.cea.pajaro@sergas.es (M.C.P.); raquel.carracedo.montero@sergas.es (R.C.M.); noelia.parajo.pazos@sergas.es (N.P.P.)

**Keywords:** resistance, urinary tract infections, *Escherichia coli*, carbapenemases, MALDI-TOF MS

## Abstract

Urinary tract infections (UTIs) are the most common infectious diseases in both communities and hospitals. With non-anatomical or functional abnormalities, UTIs are usually self-limiting, though women suffer more reinfections throughout their lives. Certainly, antibiotic treatment leads to a more rapid resolution of symptoms, but also it selects resistant uropathogens and adversely affects the gut and vaginal microbiota. As uropathogens are increasingly becoming resistant to currently available antibiotics, it could be time to explore alternative strategies for managing UTIs. Rapid identification and antimicrobial susceptibility testing (AST) allow fast and precise treatment. The objective of this study was to shorten the time of diagnosis of UTIs by combining pathogen screening through flow cytometry, microbial identification by matrix-assisted laser desorption ionisation time-of-flight mass spectrometry (MALDI-TOF MS), and the VITEK 2 system for the direct analysis of urine samples. First, we selected positive urine samples by flow cytometry using UF5000, establishing the cut-off for positive at 150 bacteria/mL. After confirming the identification using MALDI-TOF MS and filtering the urine samples for *Escherichia coli*, we directly tested the AST N388 card using VITEK 2. We tested a total of 211 *E. coli* from urine samples. Cefoxitin, ertapenem, imipenem, gentamicin, nalidixic acid, ciprofloxacin, fosfomycin, and nitrofurantoin had no major important errors (MIE), and ampicillin, cefuroxime, and tobramycin showed higher MIEs. Cefepime, imipenem, and tobramycin had no major errors (ME). Fosfomycin was the antibiotic with the most MEs. The antibiotic with the most minor errors (mE) was ceftazidime. The total categorical agreement (CA) was 97.4% with a 95% CI of (96.8–97.9)_95%_. The direct AST from the urine samples proposed here was shorter by one day, without significant loss of sensibility regarding the standard diagnosis. Therefore, we hypothesize that this method is more realistic and better suited to human antibiotic concentrations.

## 1. Introduction

Urinary tract infections (UTI) are one of the most common bacterial infections, mainly affecting adult women (50–60% lifetime incidence). The international guidelines for UTI management address limited empirical therapy for acute, uncomplicated cystitis and pyelonephritis in premenopausal women, and it is also a significant problem among elderly men with anatomical or functional urinary tract abnormalities. Empirical antimicrobial therapy should provide reliable activity against the patient’s urine organism to avoid severe complications and reduce antibiotic resistance. Therapy might be selected with attention to the likely pathogens and their anticipated antimicrobial susceptibility patterns [1] based on the susceptibility patterns of local uropathogens [1,2,3].

The diagnosis of UTIs is often based on clinical symptoms and clinical findings such as pyuria or bacteriuria, though urine culture is the gold standard for the diagnosis, but with a turnaround time of 24–48 h. Therefore, clinicians commonly need to know their local laboratory’s findings regarding antibiogram profile, especially for *Escherichia coli,* which causes around 75–95% of uncomplicated UTI episodes in women, for empirical treatments [3,4]. Direct sensitivity testing offers a rapid and accurate method to determine antimicrobial susceptibility for acute UTIs, particularly when the urine bacterial concentration is >10^5^ cfu/mL [5]. Susceptibility testing performed directly on the urine sample by disk diffusion shortens the delivery of the results by approximately 24 h, which is useful in the management of critically ill patients [6]. The direct sensitivity testing of urine has been demonstrated to be reliable in monobacterial, Gram-negative infections, decreasing the use of broad-spectrum antimicrobials [7]. Recently, more studies using automatized methods such as VITEK 2 are showing promising results [5,8,9]. Despite more studies supporting this fact, the American Society for Microbiology (ASM), the British Society for Antimicrobial Chemotherapy (BSAC), and the European Committee on Antimicrobial Susceptibility Testing (EUCAST) are skeptical of applying it because the inoculum is not standardized and it is less sensitive in detecting all bacteria present in the same sample. On the other hand, direct testing provides clinicians with early microbiological information, and permits tailored antibiotic use, and decreases antimicrobial-related adverse events [5]. 

To our knowledge, no study has yet assessed urine-specific antibiograms directly and without previous pre-processing or adjustment to the 0.5 McFarland standard. Accordingly, we compiled and analysed urine-specific antibiograms for *E. coli* using antimicrobial susceptibility testing (AST). Our goal was to prove an appropriate concordance method to determine whether the urine-specific antibiogram is a suitable surrogate for either overall or individual clinical sites. 

## 2. Materials and Methods

### 2.1. Study Design and Samples

Two hundred and eleven urine samples were selected, on separate days, over a period oessential agreement (EA), which occurs when the MIC obtained by the direct method coincides with two months during 2021 (March and April). The urine samples came from the daily clinical practice at the University Hospital Complex of Santiago de Compostela Health area. The samples obtained were transported adequately to the Clinical Microbiology Laboratory for clinical diagnostic purposes. They were pseudonymised according to ethical terms. During the mentioned period, the Clinical Microbiology Lab received a total of 6852 urine samples, of which 4756 were negatives, 844 showed contamination, and 50 presented other issues. From the 1199 positive urine samples, we randomly selected 211 following the next inclusion/exclusion criteria. 

The samples were filtered according to an inclusion/exclusion criteria, as follows:

Inclusion Criteria:-isolated identified as *Escherichia coli*-initial routine screening by Sysmex UF-5000 flow cytometer (Sysmex Corporation., Norderstedt, Germany) with more than 150 bacteria/mL

Exclusion Criteria:-non-*E. coli* isolates-samples with more than one microorganism-samples with poor or no significant growth in the cultures detected by conventional methods-samples cancelled by the VITEK 2 system for different reasons (e.g., inadequate conditions, insufficient growth in a positive control well, or technical problems, among others).

The samples selected were directly tested by the VITEK 2 system. The urine samples, which were not pre-processed, had the AST card (N-388) performed directly and were introduced into the VITEK 2 system. 

### 2.2. Flowcytometric Analysis Using Sysmex UF-5000

The Sysmex UF-5000 was used according to the manufacturer’s instructions, as given by the company (Sysmex Corporation). The Sysmex UF-5000 uses a flowcytometry-based system, with forward scatter light (FSC), side scatter light (SSC), and side fluorescent light (SFL). In addition to these components, the Sysmex UF-5000 uses depolarized side scattered light (DSS) to differentiate between red blood cells and crystals. The machine is fully automated, has a modular concept for urinalysis workflow, and can be used.

### 2.3. Urine Culture

After the initial flowcytometric analysis, all urine samples were routinely plated on chromatic biplates (CHROMID CPS Elite, bioMérieux. France). The culture plates were incubated in an ambient atmosphere at 35 °C for 18 to 24 h prior to manual interpretation. The samples were interpreted according to national guidelines [10]. Briefly, the culture-plated samples with significant growth indicative of a UTI (>10^5^ ufc/mL) contained a significant number of one or, more rarely, two significant uropathogens. The samples with non-significant growth (<10^3^ ufc/mL) that was not indicative of a UTI, or that contained three or more types of bacteria (either Gram-positive, Gram-negative, or both), were either culture-negative or mixed/contaminated, and thus did not meet the criteria used for significant urine growth. The samples with non-significant findings were not subjected to further testing. The bacteria were further identified using MALDI-TOF MS (Bruker Daltonik GmbH. Germany), followed by manual AST according to EUCAST guidelines [11]. 

### 2.4. Mass Spectrometry Identification by MALDI-TOF MS

One loop of colony, or pellet, obtained at the end of the extraction procedure was spread using a pipette tip onto the MALDI-TOF MS (Bruker Daltonik GmbH) steel plate spots and allowed to air dry. Next, one microlitre of 70% formic acid (Sigma-Aldrich, Darmstadt, Germany) was added and allowed to dry. Last, the spots were covered with the MALDI matrix (10 mg/mL α-cyano-4-hydroxy-cinnamic acid in 50% acetonitrile/0.1% trifluoroacetic acid; Bruker Daltonik GmbH). All samples were analysed in duplicate. Mass spectra were acquired in a MALDI Microflex LT/SH bench-top mass spectrometer (Bruker Daltonik GmbH) equipped with a 60 Hz nitrogen laser. FlexControl v.3.0 software (Bruker Daltonik GmbH) was used to acquire the spectra, and the MALDI Biotyper 3.1 (Bruker Daltonik GmbH) was used for real-time interpretation and identification of the microorganisms. According to the manufacturer’s instructions (Bruker Daltonik GmbH), a score of more than 2.0 indicates good species identification, a score of between 1.7 and 2.0 indicates good genus identification, and a score of <1.7 indicates unreliable identification [12].

### 2.5. Antimicrobial Susceptibility Testing (AST)

After identification, the samples were processed according to EUCAST terms [11]. Briefly, from the positive samples, a 0.5 McFarland turbidity suspension inoculum was prepared to test its susceptibility to different antibiotics by the VITEK 2 System (bioMerieux). The antibiotics tested were ampicillin, amoxicillin/clavulánic ac, cefuroxime, cefoxitin, cefotaxime, ceftazidime, cefepime, ertapenem, imipenem, gentamicin, tobramycin, nalidixic acid, ciprofloxacin, fosfomycin, nitrofurantoin, and trimetropim/sulfametoxazol (cotrimoxazole), as well as the ESBL test (N-388 AST card, bioMerieux).

The alternative method here proposed directly tested the selected urine samples by the VITEK 2 system. The urine samples, which were not pre-processed, had the AST card (N-388) directly performed and were introduced into the VITEK2 system (bioMerieux). The urine samples were not adjusted at 0.5 McFarland.

For both methods, i.e., reference and alternative, the final lecture of AST was managed by the VITEK 2 advanced expert system (AES), with a final interpretation after 7 h to 18 h, according to the manufacturer’s instructions (bioMerieux). The supervision and emission of the results were done following the laboratory’s routine procedures. 

### 2.6. Data Analysis

Descriptive statistics were used to define the samples and compare the results from the standard method to our alternative direct diagnosis. All data were analysed using Microsoft Excel and Rstudio version 3.5.0 (23 April 2018) or posterior.

For the comparative evaluation, both the degree of concordance of the minimum inhibitory concentration (MIC) values obtained by both diagnostic methods and the concordance of the clinical interpretation of these MIC values were compared. The interpretation followed the cut-off points from EUCAST [12], which established the next categories: sensitive (S), dose-dependent (I), and resistant (R).

Next, the concordances were classified into two levels [13,14]:Essential agreement (EA), which occurs when the MIC obtained by the direct method coincides with ±1 double dilution with the MIC obtained by the conventional methodCategorical agreement (CA), which occurs when the clinical interpretation of the MIC by the direct method agrees with that of the conventional method

In situations with differences between both techniques, the errors were classified as follows [13,14]:Most important error (MIE), which refers to those in which the bacteria are susceptible to the proposed method, but resistant to the reference method. They are false susceptibles, and they are the most serious in clinical practice because they involve treatment with an antibiotic that is not effective.Major error (ME), which refers to that which occurs when the bacteria being susceptible (our method classified it as resistant,) are fake resistant. This has soft implications because at no time does this harm the patient; however, a potential effective antibiotic is not used. This error is relevant when there are few therapeutic alternatives.Minor error (mE), which refers to those occurring when the antibiotic is classified as sensitive or resistant by the proposed method but intermediate by the reference method, or intermediate by the proposed method but resistant or sensitive by the reference method. They are of little importance in therapeutic decisions.

In addition, an EA and CA of ≥90% is considered acceptable and the MIE rate should be ≤3% (calculated with a minimum of 35 resistant bacteria isolates). The percentage of ME should also be ≤3% and the combined sum of ME and mE is recommended to be ≤7% [13,14]. For each CA and EA, the 95% confidence interval (CI) was estimated. For that, the exact calculation through the binomial approximation has been used, as follows:(1 − α) CI = *X* ± t_1/α_ (*s*/√n)
where *X* is the mean, *s* is the sample standard deviation, n is the size of sample, and α and t are constants. 

The CI allows us to verify whether any of the parameters studied produce significant differences in the results. The variables studied were:(1)the number of bacteria in the sample,(2)the number of leukocytes,(3)the number of leukocytes and erythrocytes,(4)the turbidity of the sample, and(5)the beta-lactamase producers.

### 2.7. Ethical Approval

The study was reviewed and approved by the Local Committee for Medical and Health Research Ethics. The need to obtain consent for the study was waived by the committee. All the urine samples were analysed for possible UTIs as ordered by the clinicians. The patient data were anonymized before flowcytometric and data analysis.

### 2.8. Limitations

The study is unique in its focus on *E. coli,* which is the main uropathogen representing more than the 80% of urine infections. More studies that include more bacteria must be performed. All the samples were monomicrobial cultures, except for the urine infections caused by two bacteria. The polymicrobial infections can represent up to 40% of urine infections, obligating us to isolate the bacteria to test the individual susceptibility. 

## 3. Results

A total of 211 samples were included in the study. All the samples were identified as *E. coli* by the Biotyper MALDI-TOF MS. For all 3376 determinations, the concordance and different errors were calculated (Table 1). Summarizing all results, the total categorical agreement (CA) was 97.4% with a 95% CI of (96.8–97.9)_95%_. The total essential agreement (EA) was also 97.4% with a 95% CI of (96.8–97.9)_95%_. The global error percentage, calculated on the total of samples analyzed, was 2.6%. The most important error (MIE) percentage was 0.5%. The percentage of major error (ME) was 1.2%. The minor error (mE) percentage was 1.0%. Cefoxitin, ertapenem, imipenem, gentamicin, nalidixic acid, ciprofloxacin, fosfomycin, and nitrofurantoin had no MIEs, and ampicillin, cefuroxime, and tobramycin showed higher MIEs. Cefepime, imipenem, and tobramycin had no MEs. Fosfomycin was the antibiotic with the most MEs. The antibiotic with the most mEs was ceftazidime.

Regarding the agreement of the results between both methods, the CA and EA exceeded 90%, which was considered acceptable, as the global agreement 97.4%. Most of the individual antibiotics showed a higher concordance, e.g., from 95% for ceftazidime to 100% for imipenem. It is worth mentioning that ceftazidime is an instable antibiotic in the face of environmental changes, which can affect susceptibility testing. 

The MIE was recalculated (rMIE) over a minimum number of resistant (non-susceptible) bacteria according to the recommendations [13,14] (Figure 1). The minimum number of resistant isolates was fixed at 35 and the acceptable rMIE less than 3%. The rMIE for each amoxicillin/clavulanic acid, nalidixic acid, ciprofloxacin, and cotrimoxazole was less than 3%. Although the rMIE for ampicillin was 3.4%, the value obtained fell into the interval of recommended confidence frame. Summarizing, the total average for all antibiotics was 1.6%, and, therefore, the rMIE was considered acceptable. Likewise, the ME was recalculated (rME) in terms of susceptible isolates (Figure 1). Three antibiotics showed values over the recommended 3%: ampicillin (3.3%), amoxicillin/clavulanic acid (4.0%), and fosfomycin (3.5%). For larger studies, this percentage will likely decrease because of the inherent sample size issue, as well as other possible variables not detected in our study.

In order to know the influence (or lack thereof) on the parameters studied, we analyzed the intrinsic parameters of the urine, such as leukocytes, erythrocytes, bacteria, or turbidity (Table 2). We found no significant differences for CI. Both CA and EA were >90%, and essentially >95%. Regarding the turbidity, previous studies used a turbidity adjusted to 0.5 McFarland, though in our study, different ranges of turbidity were measured and resulted in a CA and EA the fell over the recommended 90% and were mostly >95% for all of them. In addition, the small differences observed due to turbidity were not significant, and the confidence intervals of the concordances of the different ranges overlapped each other.

## 4. Discussion

The direct urine antibiotic susceptibility testing (duAST) of positive urine samples can reduce the diagnosis time, and therefore shorten the microbiological time of response for antibiotic profile by up to one day; as a result, the empirical use (and abuse) of antibiotics, beyond shorter overall treatment, is reduced. On the one hand, de Rosa et al. concluded that the Sysmex UF-5000 shows a high diagnostic accuracy in UTI screening with a low rate of false negatives, predicting Gram-negative bacteria with a high sensitivity and high agreement compared to cultures [15]. Li et al. suggested the combination of flow cytometry (UF 1000), MALDI-TOF, MS, and the VITEK 2 system to provide direct, rapid, and reliable identification, as well as the AST method for assessing urine samples, especially for Gram-negative bacterial infections [9]. As we have shown in our study, this algorithm could be further improved by using MALDI-TOF MS combined with duAST.

The results shown here are like other studies that used the VITEK 2 system with direct urine samples, but no one has used the original urine sample without pre-processing. Li et al. [8] reported a CA of 94.44%, Munoz-Dávila et al. [16] reported a CA of 97.6%, and Angaali et al. [9] obtained a CA of 94.3% and an EA of 97.3%. However, in all studies, the urine samples were previously processed (centrifuged, thus eliminating the supernatant), and the pellet suspension was adjusted to a 0.5 McFarland of turbidity. In all cases, the time consumption and costs were increased. The use of native urine samples is well supported because the VITEK 2 system compares the growth dynamics of the bacteria in each well with each antibiotic reading. In fact, to obtain the MICs of each antibiotic, the VITEK 2 system first standardizes the data and then neutralizes the growth characteristics of each bacterium. This is achieved by contrasting the growth parameters obtained in each well with an antibiotic with those of a positive control well. The dynamics of the model from its database with a known MIC has no influence on the comparison because each possible variable in the urine samples, such as turbidity, leukocytes, or erythrocytes, that directly affect the growth of the bacteria (pH, urine bacterial growth inhibitory proteins, or antimicrobial substances released by leukocytes) [9,16] affects the control well equally, as the wells with increasing concentrations of antibiotics. As a result, the variables do not affect the MIC results because they are all compared with the VITEK 2 system model as the trend of the bacterial growth curve in the presence of these antibiotics, and the specifics of the sample are neutralized and standardized. Therefore, we conclude that the turbidity of the samples, which differentiates this study from similar ones with direct urine samples and the VITEK 2 system, does not affect the results. Neither leukocytes nor erythrocytes, and neither the number of bacteria nor the turbidity (as already mentioned), affected the results obtained. Regardless of the values taken by the parameters, both categorical and essential agreement (CA and EA) are well above 90%, and practically always above 95%.

On the other hand, the percentage errors were as seen in previous studies (0.5, 1.2, and 1.0% for MIE, ME, and mE, respectively) where the authors adjusted the urine samples at a standardized 0.5 McFarland. Angaali et al. [10] reported errors around 0.5, 2.2, and 3% for MIE, ME, and mE, respectively, while Munoz-Dávila et al. reported a lower percentage of errors (0.2, 0.4, and 1.8%, respectively) [16]. However, these errors must be calculated over the total, and the percentage over the total, of the bacteria that are resistant (MIE) or susceptible (ME) to each antibiotic [13], and neither of the previous studies did this [9,16,17]. In our study, we calculated the errors both ways—on the one hand, to compare with the other studies (errors calculated on the global), and on the other hand, to validate our diagnostic method according to required regulations (on resistant bacteria and on sensitive bacteria). To highlight, in our study, an adjustment to 0.5 McFarland (not previously used) was performed. In addition, although Li et al. [8] obtained a rate of 1.06% for MIE, 1.59% for ME, and 2.91% for mE, they found that the MIEs were shown for imipenem, nitrofurantoin, and cotrimoxazole, while the MEs were observed for ciprofloxacin and the mEs were mainly observed for imipenem, ceftazidime, cefepime, and gentamicin. However, in our study, cefuroxime and tobramycin showed higher MIEs, while fosfomycin was the antibiotic with the most MEs and ceftazidime the antibiotic with the most mEs. Cephalosporins showed more instability than the other antibiotic families, but with not transcendence.

Finally, the concordance for the expanding spectrum betalactamasas (ESBL) was studied (Figure 2). It was greater than 90% for both the control group non-ESBL (CA = 97.8%) and the ESBL group (CA = 93.5%). Therefore, the loss of sensibility on the ESBL detection must be considered independently of CA and more associated to the detection method. However, more extensive studies must test this phenomenon because only 21 ESBL-producing bacteria were tested in this study.

## 5. Conclusions

The direct AST from urine samples proposed here was shorter by one day, without significant loss of sensibility regarding the standard diagnosis. The direct urine samples using the VITEK2 system showed good results after comparing them with the reference method (97.4% agreement). Most of the variables usually found in urine, such as cell turbidity of the sample, leukocytes, or erythrocytes, produced non-significant differences in terms of agreement with the reference method.

Importantly, the alternative method proposed here allows for targeted therapy in a shorter time than the reference method (with a difference of 24 h or more), which will have a greater impact by significantly reducing hospital stays and healthcare costs. The major limitation is the need to develop more studies with conditions closer to daily practice using more species of bacteria, more antibiotics, and rapid identification, all combined with screening techniques.

## Figures and Tables

**Figure 1 antibiotics-11-00663-f001:**
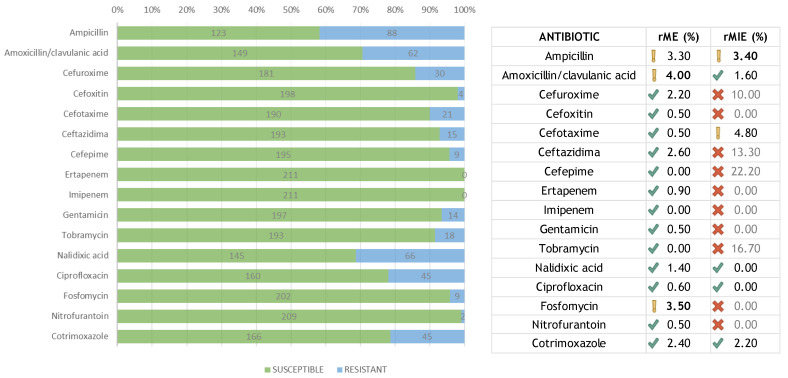
Number of susceptible and resistant isolates for each antibiotic, and their re-calculated errors.

**Figure 2 antibiotics-11-00663-f002:**
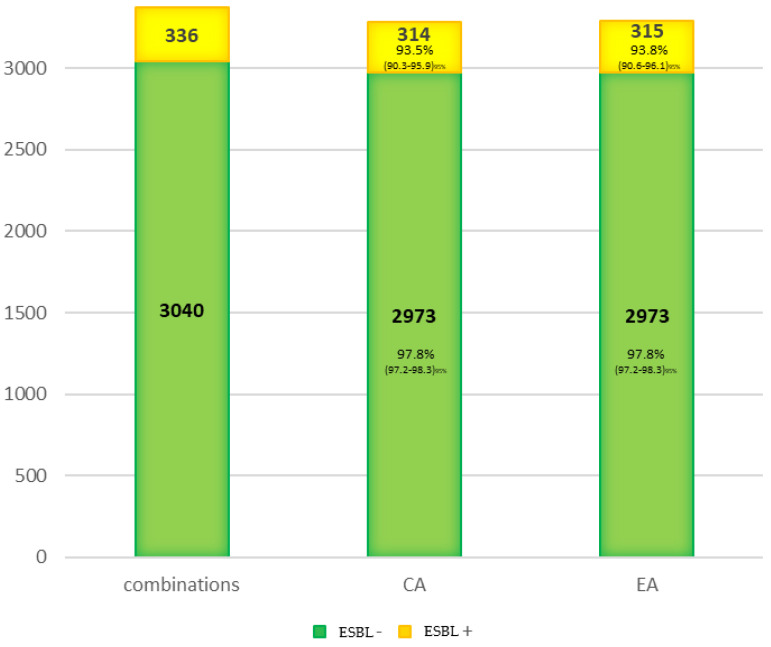
Concordance for the ESBL isolated.

**Table 1 antibiotics-11-00663-t001:** Results of the experimental method showing the errors and agreements compared to the reference method.

Antibiótic	MIE	ME	mE	CA	% CA	NO EA	EA	% EA	TOTAL
*Ampicillin*	3	4	0	204	96.7	8	203	96.2	211
*Amoxicillin/Clavulanic acid*	1	6	0	204	96.7	5	206	97.6	211
*Cefuroxime*	3	4	0	204	96.7	6	205	97.2	211
*Cefoxitin*	0	1	6	204	96.7	4	207	98.1	211
*Cefotaxime*	1	1	0	209	99.1	6	205	97.2	211
*Ceftazidime*	2	5	11	193	91.5	15	196	92.9	211
*Cefepime*	2	0	7	202	95.7	9	202	95.7	211
*Ertapenem*	0	2	1	208	98.6	3	208	98.6	211
*Imipenem*	0	0	0	211	100.0	0	211	100.0	211
*Gentamicin*	0	1	0	210	99.5	1	210	99.5	211
*Tobramycin*	3	0	1	207	98.1	3	208	98.6	211
*Nalidixic acid*	0	2	0	209	99.1	5	206	97.6	211
*Ciprofloxacin*	0	1	7	203	96.2	4	207	98.1	211
*Fosfomycin*	0	7	0	204	96.7	7	204	96.7	211
*Nitrofurantoin*	0	1	0	210	99.5	7	204	96.7	211
*Cotrimoxazole*	1	4	1	205	97.2	4	207	98.1	211
**TOTAL**	**16**	**39**	**34**	**3287**	**-**	**-**	**3289**	**-**	**3376**
**%**	**0.5**	**1.2**	**1.0**	**97.4**	**-**	**-**	**97.4**	**-**	**-**

**Table 2 antibiotics-11-00663-t002:** Concordances for (A) bacteria, (B) turbidity, and (C) leukocytes and erythrocytes. The combinations represent the number of combinations of sample/antibiotic.

**(A)**													
**BACTERIA**		**SAMPLES/Combinations**		**CA/EA**		**% CA**	**CA 95% CI**		**% EA**			**EA 95% CI**	
<5000		25/400		391/388		97.8	(95.8–99.0)		97.0			(94.8–98.4)	
5000–20000		44/704		694/692		98.6	(97.4–99.3)		98.3			(97.0–99.1)	
20000–50000		78/1248		1213/1214		97.2	(96.1–98.0)		97.3			(96.2–98.1)	
>50000		64/1024		989/994		96.6	(95.3–97.6)		97.1			(95.8–98.0)	
**(B)**													
**TURBIDITY**		**SAMPLES/Combinations**		**CA/EA**		**% CA**	**CA 95% CI**		**% EA**			**EA 95% CI**	
<0.40		25/400		391/389		97.8	(95.8–99.0)		97.3			(95.1–98.6)	
0.40–0.60		30/480		470/472		97.9	(96.2–99.0)		98.3			(96.7–99.3)	
0.60–1		54/864		845/842		97.8	(96.6–98.7)		97.5			(96.2–98.4)	
1–2		56/896		870/872		97.1	(95.8–98.1)		97.3			(96.0–98.2)	
2–3		22/352		345/344		98.0	(96.0–99.2)		97.7			(95.6–99.0)	
**(C)**													
**LEUKOCYTES**	**ERYTHROCYTES**	**SAMPLES**	**Combinations**	**CA**					**EA**				
				**Number**	**%**		**95% CI**		**Number**	**%**		**95% CI**	
					**Leukocytes**	**Erythrocytes**	**Leukocytes**	**Erythrocytes**		**Leukocytes**	**Erythrocytes**	**Leukocytes**	**Erythrocytes**
**≤** **50**	≤20	44	704	684	95.6	97.2	(94.0–96.9)	(95.7–98.3)	679	94.9	96.4	(93.2–96.3)	(94.8–97.7)
	20–100	8	128	127		99.2		(95.7–100)	126		98.4		(94.5–99.8)
	>100	1	16	15		93.8		(69.8–99.8)	15		93.8		(69.8–99.8)
**50–250**	≤20	42	672	662	95.4	98.5	(93.8–96.6)	(97.3–99.3)	663	95.6	98.7	(94.1–96.8)	(97.5–99.4)
	20–100	13	208	207		99.5		(97.4–100)	208		100		(98.2–100)
	>100	1	16	16		100		(79.4–100)	16		100		(79.4–100)
**250–1000**	≤20	29	464	444	95.8	95.7	(94.1–97.0)	(93.4–97.4)	445	96.1	95.9	(94.5–97.4)	(93.7–97.5)
	20–100	16	256	244		95.3		(92.0–97.6)	249		97.3		(94.5–98.9)
	>100	5	80	78		97.5		(91.3–99.7)	75		93.8		(86.0–97.9)
**>1000**	≤20	19	304	297	97.4	97.7	96.0–98.3)	(95.3–99.1)	296	97.6	97.4	(96.3–98.5)	(94.9–98.9)
	20–100	21	336	325		96.7		(94.2–98.4)	327		97.3		(95.0–98.8)
	>100	12	192	188		97.9		(94.8–99.4)	189		98.4		(95.5–99.7)

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
