# Peer review of "Direct Urine Resistance Detection Using VITEK 2"

_antibiotics, 2022, doi:10.3390/antibiotics11050663_

Round 1
Reviewer 1 Report
Dear authors,
I found your work interesting even if it would be more interesting, as you stated, performe a study including more bacteria.
I have some minor point to highlight:
line 1: you wrote "infectioos" instead of infectious
line 30: you wrote "no major (ME)", maybe it's better "no major error (ME)"
line 59: you wrote "promissing" instead of promising
As concern inclusion criteria: "isolated identified as E.Coli", how did you proceed to identified E.Coli? How long does it take? Can you descrive better this part?
line 146: you wrote "susceptinle" instead of susceptible and "clasiffied" instead of classified
line 154: you wrote "adittion" instead of addition
line 75: In results you state 211 sample but in paragraph 2.1 you wrote 212.
line 185-186: I don't understand the way you wrote the confidence level.
Table I: first column please check name written as title and antibiotics name.
second column title, you wrote: MIIE instead of MIE
line 198: you wrote "cetazidime" instead of ceftazidime
line 202: you wrote "aceptable" instead of acceptable
line 207: you wrote "valours" instead of values
line 215: you wrote "in other" instead of: In order
Table 2 B: you wrote "Turbidez" instead of turbidity
line 290: you wrote "such a as" instead of: such as
Author Response
Dear Reviewer
Thank you so much for the suggestions and comments. First of all, we have improved the table, and secondly, we have improved the manuscript according to the suggestions given. Specifically:
We have modified the incorrect words in lines : 1-30-59-146-154-198-202-207-215 and 290, as well as in table 1 and table 2.
We have corrected the number of samples in line 75 for two hundred eleven (211)
Also we have introduced a new section in matherial and methos where we describe in detail the identification of isolates by MALDI-TOF MS. Besides we have introduced the formula used to calculate the confidence interval (CI), and we have modified in results confidence level (which was incoreecte) by CI, too.
We hope to have reached all your comments.
Thank you so much for your suggestions
Reviewer 2 Report
The authors raise an essential topic of urinary tract infections, especially in the group of women. The study used the AST assessment using the VITEK2 system, which is successfully used in microbiological laboratories. The innovative aspect of the research was the use of this method, omitting the classic microbiological assessment, which shortened the research time. Unfortunately, as the authors themselves emphasize, this technique entails errors resulting from the presence of other elements in the urine, i.e. leukocytes. Considering the large group studied, the big mistake of this study is the lack of a statistical analysis that could confirm the possible correctness of the research performed. There are also typos and punctuation errors in the work (e.g. in tables, there are sometimes commas and sometimes dots - no uniform form).
Moreover, the figures are too laconic and incomprehensible. There are also methodological deficiencies, ie, the age group of the patients has not been explained, what symptoms they presented, etc. Most of the literature used while writing the paper is older than five years.
Author Response
Dear reviewer,
Thank you so much for the suggestions and comments. First, we have improved the manuscript according to the suggestions given. We have improved the tables according to the suggestion from the other reviewers, and secondly, we have improved the grammar and English style, now.
From our point of view, this kind of study usually do not perform a statistical study beyond of agreement and errors, that it is the way to compare two different analytical methods. We do not understand what the reviewer means with lack of statistical analysis to compare leucocites yes or not. Anyhow, the point is not to find statistical differences among non comparable groups without not control group, the study shows, first, if it is reliable the results from direct urine diagnosis and second, what antibiotic could be more affected due to the intrisic properties of urine as leukocytes or turbidity. And sorry if I missunderstanding the poit of reviewer.
Secondly, and the most importan input, this is a study comparing two analytical methods, it is not a population study, so we do not understand what is the relevance to know age, sex or illness of the population. Let me make you a question, will these data change the results and conclussion of the paper? Sorry for my trust asking directly, but we think that these data in this study are totally out of place.
Regards the references, we only show 19 refernces due to the lack of references about the field (this is the main reason to publish the paper, the novelty!), among these 19 references, 8 refences have been published during the 2018-2021 period, almost the half. The other 8 refernces are from the last decade (2011-2016), and only two refences are from 2000-2010. Then, we really apreciate if the reviewer knows more novel refernces from the last years. Thanks for that
We hope to have addressed all comments with respect to this manuscript.
Reviewer 3 Report
This manuscript offers a useful contribution to development of faster methods for detection of antimicrobial resistance in uropathogenic UTI, to more rapidly enable useful antibiotic therapy. The work is solid scientifically, and conceptually is fairly easy to follow. Its limitations are clearly indicated, but it is still relevant to treating the majority of UTIs. Its main problem is that it needs a great deal of editing for English language usage - once that is cleaned up, I think it will be a nice article.
A few comments/questions:
The flow chart provided on page 4 should be labeled with a figure number and incorporated into the manuscript - it appears to be intended to accompany the abstract?
The table shown at the top of page 7 has no caption - is it intended to be part of Figure 1? It should be integrated somehow into another table or figure, or have its own caption. At any rate, I found the relevant paragraph (lines 200-209) in the text on "recalculation" of MIE to be unclear. Without simply referring to other published work, the authors should concisely communicate the significance and method of recalculation of error rates. If it doesn't change the conclusions significantly, it may not be necessary to include.
Table 2 includes a column on "combinations" that isn't explained anywhere - please note somewhere what this refers to.
Author Response
Dear reviewer
Thank you so much for the suggestions and comments. First, we have improved the tables, and secondly, we have improved the manuscript according to the suggestions given.
We have improved the grammar and English style and reworked the ordering linking between paragraphs to allow for a more connected flow of topics and ideas for the reader.
As for the chart or diagram, certainly it has been made rathen to accompany the abstract than as a main figure.
The figure 1 is two figures, so the second figure should appear in the frame of figure 1.
Regardles the recalculation of MIE, it could be said that is a adjustment. We include the references, indeed (ref 13 and 14). On the other hand, these "recalculation" does not involve a change neither affect to the conclusions. Simply, the new parameters introduced confirm our great results. And hereby we have not extended too much through the text about it.
To end, in table 2 we have now explained what means combinations.
Thank you so much for the suggestions and comments. First, we have improved the table and the figure, and secondly, we have improved the manuscript according to the suggestions given. Specifically:
We have improved the grammar and English style and reworked the ordering linking between paragraphs to allow for a more connected flow of topics and ideas for the reader.
As for the chart or diagram, certainly has been made rathen accompany the abstract then as a main figure.
The figure 1 is two figures, so the second figure in in the frame of figure 1.
Regardles the recalculation of MIE, it could be said that is a adjustment. We includes the references, indeed (fer 13 and 14). On the other hand, these "recalculation" does not involve a change neither the conclusions. Simply, the new parameters introduced confirms our great results. And hereby we have not extend so much in the text about.
To end, in table 2 we have now explained what means combinations.
After all these modifications we have made according to the reviewer’s suggestions, we hope to have addressed all comments with respect to this manuscript.